# Parental Burden and Quality of Life in 5q-SMA Diagnosed by Newborn Screening

**DOI:** 10.3390/children9121829

**Published:** 2022-11-26

**Authors:** Heike Kölbel, Laura Modler, Astrid Blaschek, Ulrike Schara-Schmidt, Katharina Vill, Oliver Schwartz, Wolfgang Müller-Felber

**Affiliations:** 1Centre for Neuromuscular Disorders, Center for Translational Neuro and Behavioral Sciences, Department of Pediatric Neurology, University Duisburg-Essen, 45147 Essen, Germany; 2Department of Pediatric Neurology and Developmental Medicine, Hauner Children’s Hospital, LMU–University of Munich, 80337 Munich, Germany; 3Department of Pediatric Neurology, Muenster University Hospital, 48149 Münster, Germany

**Keywords:** newborn screening, quality of life, disease burden, spinal muscular atrophy, care givers

## Abstract

The aim of this study was to assess the psychosocial burden in parents of children with spinal muscular atrophy (SMA), detected by newborn screening (NBS), for which first pilot projects started in January 2018 in Germany. The survey, performed 1–2 years after children’s diagnosis of SMA via NBS, included 3 parent-related questionnaires to evaluate the psychosocial burden, quality of life (QoL)/satisfaction and work productivity and activity impairment in the families. 42/44 families, detected between January 2018 and February 2020, could be investigated. Interestingly, statistical analysis revealed a significant difference between families with children that received SMN-targeted therapy vs. children with a wait-and-see strategy as to social burden (*p* = 0.016) and personal strain/worries about the future (*p* = 0.02). However, the evaluation of QoL showed no significant differences between treated vs. untreated children. Fathers of treated children felt more negative impact regarding their productivities at work (*p* = 0.005) and more negative effects on daily activities (*p* = 0.022) than fathers of untreated children. Thus, NBS in SMA has a psychosocial impact on families, not only in terms of diagnosis but especially in terms of treatment, and triggers concerns about the future, emphasizing the need for comprehensive multidisciplinary care. Understanding the parents’ perspective allows genetic counselors and NBS programs to proactively develop a care plan for parents during the challenging time of uncertainty, anxiety, frustration, and fear of the unknown.

## 1. Introduction

Newborn screening (NBS) programs aim to achieve a presymptomatic diagnosis of treatable disorders allowing for early initiation of treatment and thus to prevent or reduce morbidity and mortality. NBS for 5q-spinal muscular atrophy (SMA) was implemented in the German public screening program in October 2021. SMA is the most common neurodegenerative disease in childhood and the second most common recessive disorder with an incidence rate of 1:6000 to 1:11,000 [1]. Homozygous deletion in the “survival motor neuron” *(SMN1)* gene, encoding the SMN protein on which motor neurons are dependent, is responsible for the disorder in more than 95% of cases [2]. There is a broad phenotypic spectrum of the disease, ranging from early onset forms being fatal in infancy if untreated, to later-onset forms that may not appear until adulthood. Today’s available therapies have led to a change in the phenotypic spectrum [3]. Milder phenotypes are associated with higher copy numbers of the most important disease modifier, a paralogous gene called *SMN2*, which is almost identical to *SMN1* but encoding for a less stable form of the SMN protein [4].

Before pharmacological treatment became available, SMA was the most frequent monogenic cause of death in infancy [5]. The spectrum of SMN-targeted therapies available includes *SMN2* splicing modifiers and gene replacement therapy that have been proven to significantly alter the course of SMA in humans [6,7,8]. Nusinersen (Spinraza^®^) was approved as the first drug for SMA in the USA in December 2016 and in the EU in July 2017. An antisense oligonucleotide drug, Nusinersen modulates pre-messenger RNA splicing of the *SMN2* gene. Intrathecal administration with Nusinersen via injection is performed on treatment days 1, 15, 30, 60, 180 and then every 120 days; dosage is 12 mg for all children independent of age [9]. Onasemnogene Abeparvovec (Zolgensma^®^) is an adeno-associated viral vector-based gene therapy designed to deliver a sound copy of the *SMN1* gene to the motor neurons through a single intravenous infusion [8] and was approved in the EU in July 2020. Ongoing monitoring is needed from all sources of safety data including hepatotoxicity, thrombocytopenia, cardiac adverse events and events suggestive of neuronopathy [10].

Pilot projects [11] starting in January 2018 in Germany showed a significant improvement in diagnostic delay and consequently in motor outcomes in early treated patients with subsumable severe forms [12,13]. In SMA with four or more *SMN2* copies, treatment was initially postponed in the pilot projects (“wait-and-see strategy”) because the time of onset of the disease was unclear.

While feasibility, accuracy, and clinical utility of NBS for SMA are meanwhile unquestioned, there is still a lack of knowledge about the psychosocial impact of NBS on families. We hypothesize that a substantial degree of uncertainty, e.g., due to the lack of long-term data in treated patients and/or a “watchful waiting” strategy in patients with 4 *SMN2* copies are the main reasons for psychosocial stress in SMA-NBS families. This in turn defines (i) the need of a family-focused approach aiming to evaluate the need for psychosocial support for parents and (ii) the necessity to better understand the families’ perspectives on NBS programs. To address the need of further studies related to SMA, we performed a study based on three questionnaires one year after diagnosis through NBS aiming to investigate the burden of this program in terms of an early SMA-diagnosis on families.

## 2. Materials and Methods

### 2.1. Study Protocol

Inclusion criteria for this study was to be a parent of a newborn who was confirmed positive for SMA in the German pilot projects and born between 15 January 2018 and 1 March 2020.

Screening for SMA was initially performed as part of a pilot project based on genetic screening for cystinosis and SMA [11]. Since May 2019, screening for SMA has been continued on the co-authors’ initiative of this work. Details of design, ethics, treatment, assessments, and a discussion on burden of parents with children with 4 *SMN2* copies were published previously [12,14,15].

Treatment protocol of the first two years of the pilot projects provided for a treatment decision by the recommendations of the “American SMA-NBS Multidisciplinary Working Group” was published in 2018 [16]: Immediate treatment with Nusinersen was recommended for children with 2 and 3 *SMN2* copies and a “watchful waiting” strategy to children with ≥4 copies. Every 2–4 months, patients underwent regular standardized neuropediatric examination, comprising electrophysiological examination and standardized physiotherapeutic examination [13]. Treatment by that time was initiated solely with Nusinersen, as Onasemnogene Abeparvovec was not available in Germany until 2021. Psychosocial evaluation was started 12 months after NBS at the earliest, at a time when the children had already gone through the major motor developmental milestones and when no temporally immediate effects of the diagnosis notification on the parents could be assumed.

### 2.2. Caregiver Outcome Measures

Caregiver/parent outcomes were obtained using self-administered questionnaires. Two questionnaires, extensively validated for families with chronically ill children living in Germany, were chosen to measure quality of live and burden of disease: the “Family Burden Questionnaire” (FaBel) [17] and the “Quality-of-Life Inventory for Parents of Chronically Ill Children” (ULQIE). To assess the impact of a disease on a patient’s ability to work and/or to perform non-work activities, the “Work Productivity and Activity Impairment” (WPAI) [18] questionnaire was used. The paper-based questionnaires were distributed by mail after patients had been informed about the survey via phone call or in the setting of a routine-follow-up appointment. All questionnaires were written in German and all parents were able to read and understand German; due to the small sample size, patients were solely stratified into “treated” and “untreated/watchful waiting strategy” patients.

The “FaBel” is a German version of the Impact on Family Scale [19] that measures the burden of a chronic illness of a child on the patient’s family. The summary scale ranges from 0 to 4, while higher scores indicate a greater family burden. The questionnaire contains 33 Likert-scaled items to assess the generally negative impact of the disease on parents, the description of social relationships, the concern for siblings, the financial impact and problems in coping as well as a total score.

The “*ULQIE*” is a questionnaire designed according to the principles of classical test theory. The subscales are developed by factor analysis. The 29-item instrument contains the dimensions of physical and daily social functioning, satisfaction with the situation in the family, emotional distress, self-development, and well-being [20].

The “WPAI” questionnaire was used in a caregiver version (WPAI-CG) in which the effect of a child’s specific health problem on the parent’s work productivity was measured. In this questionnaire higher scores indicate a higher level of burden.

### 2.3. Analysis

The questionnaires were recorded in an anonymized data mask. Statistical analyses were performed using IBM SPSS Statistics 27. Descriptive statistics were used to summarize sociodemographic information. Means and standard deviations were used for continuous variables, and differences in means were analyzed using t-tests for independent samples. Frequencies and proportions were used for categorical data. Analysis of variance (ANOVA) was also used, and the statistical test was two-sided with a significance level of 0.05.

## 3. Results

### 3.1. Participation

A total of 45 SMA patients, originating from 44 families, were identified via newborn screening between January 2018 and February 2020. Two untreated children with 2 *SMN2* copies died at the age of 5 months and consequently their parents were not contacted for participation. Out of the remaining 42 families, at least one parent of each of the screen-positive newborns participated, and for parents with one response, the non-participant indicated they were in complete agreement with their partner and did not complete a separate survey. 38 mothers and 35 fathers participated, thus a total of 73 surveys could be included.

### 3.2. Characteristics and Motor Development of Infantile SMA Patients

39.5% (17 patients) had 2 *SMN2* copies, 21% (10 patients) had 3 *SMN2* copies, and 39.5% (16 patients) had ≥4 *SMN2* copies [13]. Details on the motor outcome of the patients from the pilot projects were previously published by the NBS study group [12,13,21]. Figure 1 shows the children of the study cohort with 2 *SMN2* copies, treated as soon as possible after diagnosis confirmed and at least 12 months of age. This cohort was suspected to develop the clinical phenotype of SMA type 1 or 2 if left untreated. The majority of early symptomatic patients, defined by symptoms at birth or development of symptoms in the first 4 weeks of life, showed a delayed motor development. One patient developed disturbed motor development with swallowing difficulties and respiratory insufficiency. All clinically asymptomatic patients developed motor milestones in time, which was not expected in the natural disease course of SMA type 1 or 2.

### 3.3. Family Burden Questionnaire (FaBel)

A total of 51/53 questionnaires were feasible for evaluation, 42 primary caregivers being most responsible for the care, and 9 non-primary caregivers completed the form. All primary caregivers in the study were the mothers of the children. Mothers and fathers showed no differences in the descriptive analysis. The highest score of burden was obtained in the item “personal strain/worries about the future” (mean 2.35/SD 0.60). The analysis of families with treated children vs. untreated children revealed a higher score for the families with treated children in the descriptive analysis (Figure 2). The statistical analysis showed a significant difference between families with treated vs. untreated children in the items “social burden” (*p* = 0.016) and “personal strain/worries about the future” (*p* = 0.020). A statistical trend could be seen in the burden of the sibling (*p* = 0.055) and the total score of burden (*p* = 0.056).

### 3.4. Quality-of-Life Inventory for Parents of Chronically Ill Children (ULQIE)

We conducted a statistic analysis of the questionnaire as to physical and daily social functioning, satisfaction with the situation in the family, emotional distress, self-development, and well-being. A total of 73 questionnaires were feasible for evaluation (38 mothers/35 fathers), 11 questionnaires could not be evaluated because of too many missing items. In total, the parents showed only moderate limitations in QoL. Most noticeable was the restriction in the item “self-development” (mean 1.80/SD 0.79). In addition, we did not find a statistically significant difference between fathers and mothers. There was no difference between parents with treated and untreated children (Figure 3).

### 3.5. Work Productivity and Activity Impairment-Care Givers (WPAI-CG)

During the study, the family situations underwent some changes due to the COVID-19 pandemic. The majority of parents worked from home so the WPAI questionnaire did not fit the situation of all families, thus we relinquished an in-depth statistic evaluation. Out of 42 families, 38 fathers and 35 mothers answered the questionnaire, including 35 fathers (working), 3 fathers (non-working), 12 mothers (working), and 23 mothers (non-working). Due to a large number of unemployed women, the sample size of working mothers was too small to be analyzed. In the group of fathers, the WPAI showed significant results. The fathers of treated children felt more negative effects on their productivities at work and showed more negative effects on daily activities (*p* = 0.022) in the last seven days than fathers of untreated children (*p* = 0.005). The results for the total group are similar to the group of fathers due to the high prevalence of working fathers.

## 4. Discussion

We present data on the psychosocial burden and quality of life in a large cohort of families with SMA patients detected by a newborn screening project for SMA in Germany. Despite the undisputed clinical benefits of SMA-NBS on motor development (Figure 1), several challenges have been noted, one of which being the potential psychological impact on the child’s family. Concerns about the potential for psychosocial disturbances to familial bonds from other NBS programs have focused on parents receiving false-positive NBS results [22]. However, although there are different studies thus far that have prospectively explored the psychosocial burden of families with genetically confirmed screening results after NBS [23,24] as well in neuromuscular disorders like Pompe disease [25,26], there is only one study focusing on this topic in SMA in Australia with a good acceptance for a pilot project from a parent perspective [27]. Thus, we performed the first study investigating the psychosocial impact of early diagnosis via genetic NBS for SMA, with the purpose to improve the follow-up care of families after NBS.

A cohort of 42 families answered three questionnaires one year after diagnosis of SMA and provided data on the long-term impact on parents’ health and well-being. The most important findings in this cohort were the high scores in “personal strain/worries about the future” in all families. Interestingly, they were significantly higher in the group of parents whose children were treated as compared to those with a “watchful waiting” strategy. This result is similar to parents with chronically sick children as reported in the literature before [17,28]. However, we cannot exclude a possible bias of this result with respect to *SMN2* copy number in this study setting. The only patients offered a wait-and-see strategy were those with 4 *SMN2* copies. Because the natural history of these SMA patients is generally milder than in the forms with 2 or 3 *SMN2* copies, the lower future concerns might also be related to the higher copy number. However, there were no differences between families with children with 2 and 3 *SMN2* copies, which contradicts this assumption. In addition, the survey was conducted at a time when the good response of the children to the medication with mostly even normal development and the achievement of milestones was already clearly visible to the parents. Thus, we speculate that the initiation of treatment might change the parents’ perception of being threatened by disease, whereas the “watchful waiting” strategy may raise the hope that laboratory/molecular genetic findings do not necessarily mean a manifestation of the disease. Similar results were found in newborn screening for metabolic disorders. On the one hand, early diagnosis and treatment will lead to favorable physical and cognitive outcomes, on the other hand, dietary treatment and diagnoses bearing risk for metabolic decompensation despite treatment are associated with a higher perceived burden for the family [29].

In this cohort, only a few children were severely affected. Nevertheless, the existing desire of many families to change the treatment strategy from Nusinersen, a continuous therapy on RNA level, to Onasemnogene Abeparvovec, a “one-time” therapy on the DNA level, could also be interpreted as an indication in this direction. Figure 1 shows (here in patients with 2 SMN2 copies) that some families decided to change therapy despite very good motor development of their children. This could be interpreted as a desire for a kind of “final solution to the problem”, which is hoped to reduce the permanent stress of a chronic disease with a chronic need for medication. In the end, we should prepare the parents from the first appointment, that SMA-NBS will change the natural course of the disease to become a treatable chronical disease. However, Rodrigues and colleagues reported before that Nusinersen treatment did not impact proxy-reported QoL in children with SMA, whereas gastrostomy tube and ventilation support decreased children’s QoL [30].

Another significant and very interesting result was the high score for the item “social burden” (*p* = 0.016). Families feared negative effects of the disease on their peer group, loss of employment, and lack of participation in social life (e.g., going to restaurants, vacations, family celebrations). In light of this, one could speculate that patients with NBS-detected disorders are still perceived by their parents as chronically ill children, even if they show almost normal motor development under treatment. In a review of QoL in SMA patients, the authors concluded that the analysis of parents’ questionnaires revealed that different types of SMA and clinical treatment can significantly affect the QoL of SMA patients [31]: the more severe the SMA manifestation, the lower the average scores on the PedsQL NMM (Neuromuscular Module) and PedsQL FIM (Family Impact Module) as reported by the patients’ caregivers, and the poorer was the patients’ QoL [31]. Studies of a large Chinese cohort revealed that disease-related clinical features and clinical treatments have a significant impact on QoL of patients with SMA. Particularly, QoL was relatively poor in children with type I and type II SMA as well as in their caregivers compared to those with SMA type III [32]. Further, the clinical relevance of reported disturbances of QoL in caregivers should support the implementation of adequate support services for families of symptomatic SMA patients [33].

In contrast, analysis of QoL in all families in this study after SMA-NBS revealed results similar to other families with chronically diseased children (diabetes mellitus and epilepsy) [34] with no differences in the mean values between treated vs. untreated children. The difference of the better QoL could be explained by the unexpected achievement of motor milestones after NBS and early treatment.

WPAI-CG showed significant negative effects on productivity at work and also more negative effects on daily activities of the parents. These findings build upon former literature showing a substantial burden among caregivers of pediatric patients with chronic neurological diseases [35]. Of note, WPAI-CG scores were higher among caregivers of chronically ill children (6 months–11 years of age) and adolescents (12–17 years of age) relative to caregivers of chronically ill adults (at least 18 years of age). Specifically, parents of chronically ill children had the highest mean percentage of overall productivity loss (55%) and lowest employment rates (33%) [36]. Our survey revealed that 23 mothers were not employed more than one year after their child’s SMA diagnosis in NBS, in contrast to only 3 fathers who were not employed. This observation is consistent with the general situation of families in Germany in 2019: Only 8.9% of mothers are still employed after the birth of a child (German Federal Statistical Office). In Germany, families receive financial support in the first year after birth if one of the care-givers stays at home. However, further development of the parents’ employment status after the first year in SMA-NBS should be re-evaluated in future in order not to overlook the financial stressor that is omitted from the forgone family employment (FFE) because of the child’s health status. FFE was defined as any family member who stopped working and/or reduced their hours because of the child’s health or medical condition. This is the primary factor in financial toxicity and a burden for families with chronically ill children [37].

To sum up, this study revealed the high disease burden experienced by caregivers after SMA-NBS. Notably, the impact was manifested in a variety of functional, emotional, and social elements of caregivers’ lives, and extended beyond the acute time interval of diagnostic and initiation of treatment. We strongly recommend foresightful psychosocial counseling not only to support the parents to overcome the stress of diagnosis and choosing the right medical treatment in the first appointments but especially in the following years to arrange with the chronic condition of their child.

## 5. Conclusions

The study shows that the diagnosis of SMA after NBS has serious implications for families, regardless of the improved neurological outcomes of children after timely medical treatment. There is no doubt that NBS will dramatically improve the prognosis of children with SMA. Although the sample size in our study was small, the results should be included in the current expended national-wide NBS program, our data show clearly that the care of these patients and their families must include a multidisciplinary approach to address psychosocial problems in addition to neurological problems. Long-term data on quality of life, psychosocial distress, and their impact on child development need to be studied in the future.

Nevertheless, the new treatment options have uncertain efficacy and long-term side effects, so parents need professional guidance from an expert team in a specialized medical center.

### Limitation of the Study

An in-depth statistical evaluation of the data set between the various medical treatment groups or dependency on SMN2 copy numbers could not be performed due to the small sample size. However, a limitation of NBS for SMA is its inability to securely distinguish SMA type 1 from SMA type 2 or 3 via *SMN2* copy numbers.

## Figures and Tables

**Figure 1 children-09-01829-f001:**
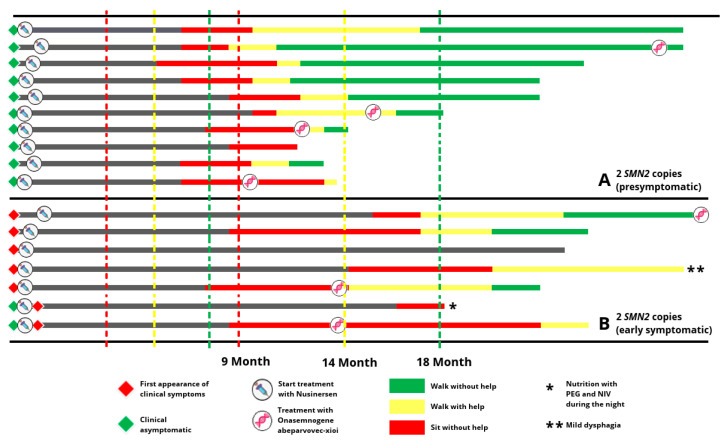
Motor development of SMA patients with 2 *SMN2* copies identified through newborn screening showed almost normal development at the time of evaluation, in particular in asymptomatic patients at the start of treatment.

**Figure 2 children-09-01829-f002:**
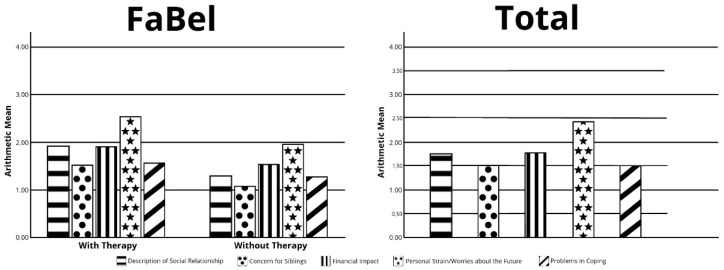
FaBel: Evaluation of burden in families after NBS, A: primary caregivers, B: total score, in families with treatment (Nusinersen/Onasemnogene Abeparvovec) and without treatment (watchful waiting).

**Figure 3 children-09-01829-f003:**
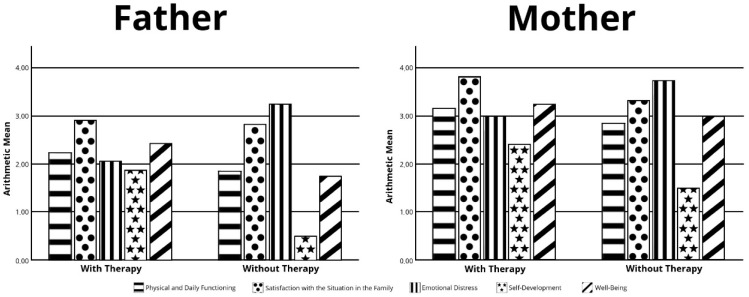
ULQIE: Evaluation of QoL in families with treatment (Nusinersen/Onasemnogene Abeparvovec) and without treatment (watchful waiting) A = father, B = mother.

## Data Availability

The data supporting the findings of this study are available from the corresponding author upon reasonable request.

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
