# Peer review of "Parental Burden and Quality of Life in 5q-SMA Diagnosed by Newborn Screening"

_children, 2022, doi:10.3390/children9121829_

Round 1

Reviewer 1 Report

In this article, the authors use a battery of family-centric surveys to ascertain the effects of newborn screening and treatment/monitoring for SMA have on nuclear families. Notably and somewhat counterintuitively, the perceived psychosocial stressors of the disease were more profoundly noted in families of treated children versus those who had conservative monitoring temporally.  This article is well-written with logical flow and a nice introduction to SMA both molecularly and regarding the psychosocial aspects. Notably, where the paper would be greatly improved is in the quality of the figures. In their current state, they are blurry and completely illegible, rendering independent interpretation of them impossible. 

Additionally, while the work is interesting and similar studies have not been published, it is unclear how generalizable these data are to the broader population. Could some of the stressors be due to single parent income given that only 9% of women work postpartum in Germany? Were any analyses done to combine data for single parents rather than just single fathers? The discussion feels underdeveloped at this point and could stand to discuss some of these factors regarding broader applicability, future directions and how this information should guide the expansion of NBS as SMA becomes a more treatable, chronic disease? Should families receive more upfront support with their initial visit to review results? 

Finally, the discussion ends very abruptly with no real concluding statement and also has cut and paste directions from within the instructions to authors which needs to be rectified. 

Minor concerns: Line 215 has type and should read "purpose."

There is inconsistency between the use of commas and periods for decimal places. 

Author Response

Reviewer 1:

In this article, the authors use a battery of family-centric surveys to ascertain the effects of newborn screening and treatment/monitoring for SMA have on nuclear families. Notably and somewhat counterintuitively, the perceived psychosocial stressors of the disease were more profoundly noted in families of treated children versus those who had conservative monitoring temporally.  This article is well-written with logical flow and a nice introduction to SMA both molecularly and regarding the psychosocial aspects.

  1. Notably, where the paper would be greatly improved is in the quality of the figures. In their current state, they are blurry and completely illegible, rendering independent interpretation of them impossible. 

Reply: Thanks for this important remark, all figures have been uploaded as a TIFF file.

  1. Additionally, while the work is interesting and similar studies have not been published, it is unclear how generalizable these data are to the broader population. Could some of the stressors be due to single parent income given that only 9 % of women work postpartum in Germany?

Reply: Thanks for your comment. The German government supports families financially in the first year after birth to make sure that one of the parents can stay at home to take care of the baby, but after this period, especially when the baby has health problems, the financial burden of the families will grow. This aspect should be discussed early with the parents to make sure that finical stress is not a big point in the decision for a treatment option. We have added this point in the discussion section (page 7, lines 275–282 and lines 286–290).

  1. Were any analyses done to combine data for single parents rather than just single fathers?

Reply: Thanks for this annotation. The analysis of the total group showed similar results as the analysis of single fathers, because the number of fathers is significantly higher than the numbers of mothers. We have added this point in the result section (page 6, lines 203–204).

  1. The discussion feels underdeveloped at this point and could stand to discuss some of these factors regarding broader applicability, future directions and how this information should guide the expansion of NBS as SMA becomes a more treatable, chronic disease?

Reply: Thanks for this remark. We have extended the discussion section including future direction for families with SMA babies. Today, SMA is a treatable disease, but regardless of all our medical efforts, most of the babies will develop symptoms over time and this must be discussed with the parents from the first appointment. We have added these points in the discussion section (page 6, lines 248–250) and in the conclusion section (page 7/8, lines 305–313).

  1. Should families receive more upfront support with their initial visit to review results? 

Reply: Thanks for this remark which is simple to answer: Yes. We have extended the summary for this important point (page 7, lines 297–298).

  1. Finally, the discussion ends very abruptly with no real concluding statement and also has cut and paste directions from within the instructions to authors which needs to be rectified. 

Reply: Thanks for this valuable advice, we have cut the directions from the journal and extended the discussion section by including a sum-up statement (page 7; lines 294–300).

  1. Minor concerns: Line 215 has type and should read "purpose."

Reply: Thanks for this note, we have corrected the typo.

  1. There is inconsistency between the use of commas and periods for decimal places. 

Reply: Thanks for this note, we have corrected this inconsistency.

Reviewer 2 Report

This paper is well written, interesting to the general readership, with new knowledge. Methodologically correctly set, clear goals, the discussion supports the results with quality, and the references correctly follow the entire article.

Minor language corrections are required.

Author Response

Reviewer 2: This paper is well written, interesting to the general readership, with new knowledge. Methodologically correctly set, clear goals, the discussion supports the results with quality, and the references correctly follow the entire article.

Reply: Thanks for this positive estimation of our work.

Minor language corrections are required.

Reply: Thanks for this note, we have performed a professional language correction.

Round 2

Reviewer 1 Report

The authors have adequately addressed all my concerns at this time and the revisions are well-written and ready for publication from my perspective. Thank you for this interesting work!